

# DeepCorr: a novel error correction method for 3GS long reads based on deep learning

Rongshu Wang and Jianhua Chen

Department of Electronic Engineering, Information School, Yunnan University, Kunming, Yunnan, China

## ABSTRACT

Long reads generated by third-generation sequencing (3GS) technologies are involved in many biological analyses and play a vital role due to their ultra-long read length. However, the high error rate affects the downstream process. DeepCorr, a novel error correction algorithm for data from both PacBio and ONT platforms based on deep learning is proposed. The core algorithm adopts a recurrent neural network to capture the long-term dependencies in the long reads to convert the problem of long-read error correction to a multi-classification task. It first aligns the high-precision short reads to long reads to generate the corresponding feature vectors and labels, then feeds these vectors to the neural network, and finally trains the model for prediction and error correction. DeepCorr produces untrimmed corrected long reads and improves the alignment identity while maintaining the length advantage. It can capture and make full use of the dependencies to polish those bases that are not aligned by any short read. DeepCorr achieves better performance than that of the state-of-the-art error correction methods on real-world PacBio and ONT benchmark data sets and consumes fewer computing resources. It is a comprehensive deep learning-based tool that enables one to correct long reads accurately.

## INTRODUCTION

Next generation sequencing technologies produce accurate but short reads, with a maximum read length of approximately 600 bases, which makes it very complicated in subsequent reconstruction and analysis (*Treangen & Salzberg, 2012*). The third-generation sequencing technologies represented by PacBio and Nanopore platforms can produce long reads up to 10∼15 kbp, which provides a chance to solve challenging downstream problems such as *de novo* assembly (*Roux et al., 2019*), variant calling (*Zojer et al., 2017*; *Wang & Chen, 2023*). Among them, the error rates of PacBio reads are high (∼13%). Although this technology has been improved greatly in recent years due to the new sequencing chemistries and the sequencer, the error rates of "single-pass" reads also known as continuous long reads (CLR, with length >25k) are remained the same as the beginning ∼13% (*Quail et al., 2012*). Then PacBio releases a circular consensus sequencing (CCS, with length 10∼20k) template in 2021 allowing to sequence molecules around several times to maximize read accuracy (99.99%) (*Foord et al., 2023*). Nanopore technology cannot sequence the same

Corresponding author
Jianhua Chen, chenjh@ynu.edu.cn

molecule multiple times as PacBio, and the error rates of Nanopore reads are ∼15% (*Li et al., 2016*). Although the error rate is not satisfactory, the two platforms have accumulated a large amount of long-read data in the past ten years. Thus, there is still a great commercial advantage to algorithmically manage the high error rates of the accumulated longer, "single-pass" reads from both mainstream platforms. Therefore, combining cost-effective homologous high-precision short reads to polish and correct long reads is an economical way, which has successfully aroused the interest of many researchers (*Ye & Ma, 2016*).

As mentioned in the research survey, the long-read error correction algorithm is divided into self-correction algorithms and hybrid correction algorithms (*Morisse, Lecroq & Lefebvre, 2020*). Self-correction algorithms, such as CONSENT (*Morisse et al., 2021*) and the latest VeChat (*Luo, Kang & Schönhuth, 2022*) are purely based on the information contained in the long reads. As a result, deeper long read coverages are usually required, and self-correction can thus prove to be inefficient when dealing with datasets displaying low coverages (*Sedlazeck et al., 2018*). In contrast, hybrid correction utilizes complementary, high-quality short reads for correction. A key advantage of hybrid correction is that error correction is primarily guided by the short-read data. Therefore, the sequencing depth of the long reads does not affect this strategy in any way. The hybrid error correction algorithm can be further categorized into four types: short-read alignment-based, short-read assembly based, De Bruijn graph (DBG)-based, and hidden Markov model (HMM)-based. PacBioToCA (*Au et al., 2012*), Proovread (*Hackl et al., 2014*), Nanocorr (*Goodwin et al., 2015*), ColorMap (*Haghshenas et al., 2016*), *etc.*, are short-read alignment-based methods. These methods try to get the consensus between short reads and the aligned fragments of a long read with different strategies. Short-read assembly-based methods, such as ECTools (*Lee et al., 2014*), HALC (*Bao & Lan, 2017*), and MiRCA (*Kchouk & Elloumi, 2016*), assemble short reads into longer contigs in advance. In this way, the assembled contigs can be effectively aligned to repetitive and noisy regions in long reads. LoRDEC (*Salmela & Rivals, 2014*), Jabba (*Miclotte et al., 2016*), and FMLRC (*Wang et al., 2018*) are based on another strategy referred as DBG. These methods construct DBG with high frequency short-read k-mers, then anchor the long reads to the DBG and traverse the graph to obtain an optimal path. Most of the above methods are designed based on the error profile of the reads from different platforms. For example, PacBioToCA, Proovread and LorDEC, *etc.*, are for PacBio data, and Nanocorr is for Nanopore data. This means that such an algorithm is sequencing-technology-dependent. At the same time, many methods such as Jabba and Proovread report trimmed corrected reads, which leads to a certain degree of loss of the length advantage of original long reads. Therefore, it would be perfect if the method can ensure the corrected long reads are untrimmed and available to reads from both platforms.

Machine learning is widely used in genome sequence analysis due to its ability to make objective automated decisions (*Javed et al., 2023*). Based on these, Hercules (*Firtina et al., 2018*) models each complete long read as an HMM, and refine the parameters based on the error profile of any other error-prone sequencing technology, including PacBio and Nanopore. Hercules is the only machine learning-based method and is very good at attaining the short-term dependencies among neighboring regions in a sequence

(*Cao et al., 2024*). However, HMM has certain limitations in capturing long-term dependencies because it is usually based on the assumption of a finite state space and its ability is limited by the number of orders and parameters of the model (*Durbin et al., 1998*). As a result, the short-term dependencies trained by high-precision short reads in Hercules can only repair the regions aligned with short reads, which is almost ineffective for the remaining unaligned regions. In addition, the training of HMMs is quite time-consuming.

At the same time, artificial neural network algorithms have been found to be able to effectively capture and process long-term dependencies between sequences than HMM (*Durbin et al., 1998*; *Salaün, Petetin & Desbouvries, 2019*), and have already been applied to improve the HMM-solved genomics research such as prediction of nucleosome positioning (*Gòdia et al., 2023*), base calling (*Boza, Brejova & Vinar, 2017*), and gene expression inference (*Machado et al., 2023*).

DeepCorr, a hybrid error correction method based on RNNs is proposed. In this work: (1) each complete long-read sequence is treated as a time series, and no trimming is applied to this sequence; (2) the recurrent neural network is adopted for the first time in error correction to capture the dependencies between bases. This innovative approach not only corrects the aligned regions of long reads but also take good care of those bases in the uncovered regions (called gaps), which rarely handled by previous alignment-based error correction methods. (3) the method is sequencing-technology-independent and can ignore the error profile of different sequencing platforms. The model consists of a bidirectional gated recurrent unit (Bi-GRU) (*Chung et al., 2014*) layer and a *softmax* layer. The homologous high-precision short reads are aligned to the long reads to generate the features and labels for model training. The cross-entropy function is the loss function for back-propagation (*Zou et al., 2019*). In the final step, the long reads to be corrected are fed into the well-trained model to predict the most probable base at each position. In this way, the task of long-read error correction is transformed into a base classification challenge at each genomic position.

We evaluate the method on benchmark datasets proposed by LRECE (*Zhang, Jain & Aluru, 2020*), and the experimental results show that compared with the only machine learning-based method Hercules, DeepCorr has a higher alignment identity and more aligned bases, indicating that the unaligned bases are taken into consideration carefully. In addition, DeepCorr gets the highest alignment identity while maintaining the read lengths and performs better than other non-machine learning-based error correction methods. In addition, the no-trimming policy can ensure that the N50 indicator and length advantages not be affected, which will facilitate the downstream *de novo* assembly. In conclusion, it is a comprehensive and balanced deep learning-based method for long-read correction.

## MATERIALS AND METHODS

Portions of this text were previously published as part of a preprint (*Wang & Chen, 2023*).

The proposed algorithm includes absolute position generation, feature vectors and label generation, model training and base prediction steps. Assuming that the length of the long read to be corrected is $T$, the input of the network should be the feature vector of each

**Figure 1** Generation of absolute positions. "Position" is the original position of the long-read base, and "Absolute Position" is the position in the sequence after taking into consideration of inserted bases. The gray blocks indicate the short-read bases are the same as those in the long read, where mismatched bases are specifically marked. The dash in the long read indicates an insertion at that position, and the dash in the aligned short reads means a deletion at that position. In the example, there are five short reads aligned with the current long read from position 1 to position 31. Because a base "T" is inserted behind position 3 for all five short reads, one position is added for the inserted base. Two bases "GC" are inserted behind position 10 for all five short reads and two positions are added in the long read. Except for the corresponding number of positions added to the inserted base, the positions of the following bases are updated accordingly. After the absolute position is generated, the feature vector and label of each position will be generated based on these absolute positions.

position $X_1, X_2, \ldots, X_T$. The model predicts the target value $Y_1, Y_2, \ldots, Y_T$ of each position according to the feature vector, and the base sequence can be obtained according to the predicted target value to finish the correction.

## Absolute position generation

The basic RNN-based classification framework requires that the input sequence and the output sequence have the same length, that is, the number of feature vectors and the number of labels are the same, and they should be in one-to-one correspondence. To find and fix this correspondence during error correction, the alignment tool Minimap2 (*Li, 2018*) is used to align the short reads to the long reads to be corrected to generate the alignment information. From the alignment information, not only can we count the number of times the current base in the long read is covered by all kinds of bases in the aligned short reads but also the details of insertions or deletions. However, after observing the alignment information, it is found that if the feature vector of each long-read base is directly generated when insertions or deletions occur in any position of the long read, the one-to-one correspondence between a feature vector and a label will be damaged. To overcome the obstacle, one position for each inserted base will be reserved, and the absolute position information of each base in the sequence is updated. In fact, the alignment information of each base in the long read is browsed first to find the positions where insertions occur, and then the positions in proper numbers are reserved for each of these positions. If only deletion occurs, the position of the current base is kept as it is. With the absolute position information of the bases, the corresponding relationships between feature vectors and labels can be constructed and kept. The absolute position generation is shown in Fig. 1.

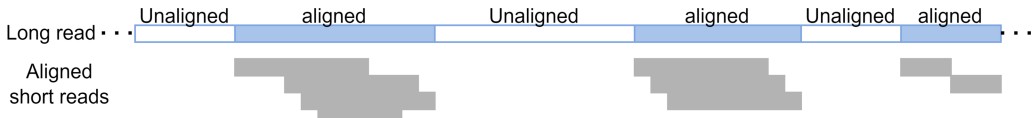

**Figure 2  Long read coverage.** The transparent bars represent the regions on the long reads that are not covered by the short reads, which are surrounded by aligned regions both forward and backward.

**Table 1  Percentage of long reads aligned by short reads.**

| Dataset | *E. coli* PB | *E. coli* ONT | Yeast PB | Yeast ONT | Fruit fly PB | Fruit fly ONT |
|---------|-----------|------------|---------|----------|-------------|--------------|
| Covered | 6.38% | 3.36% | 2.86% | 13.94% | 24.94% | 9.79% |

## Feature vectors and label generation

By observing the coverage of long reads after alignment, it is found that the regions on the long read that are not aligned with any short read are generally short and discontinuous, that means each unaligned region is surrounded by aligned regions, as shown in Fig. 2.

Therefore, we attempted to obtain the proportion of bases in long reads that are aligned with short reads, as shown in Table 1. Specifically, we use long reads as the target sequences and short reads as the query sequences. Minimap2 is employed to align the query to the target, and the commonly recognized third-party statistical tool *pysam* is used to retrieve the coverage data. The tool *pysam.AlignmentFile.pileupnsegments* is used to obtain the number of short-read coverages at each position on the long reads from the SAM (*Li et al., 2009*) files. We calculate the proportion of the number of positions with coverage>0 to the total number of long-read bases. After statistics, most of the positions on the long reads of the six benchmark datasets from LRECE (*Zhang, Jain & Aluru, 2020*) are not covered by any short reads.

These uncovered positions are considered to have lost the basis for error correction due to the lack of short-read coverage, and the previous alignment-based error correction algorithms such as PacBioToCA, cannot handle these positions well. On the other hand, Proovread cuts each uncovered region in half iteratively until it could be queried reversely to find short-read fragments that could be aligned with this uncovered small fragment, which will produce too many false positive anchor points and destroy the continuity of long reads (*Brudno et al., 2003*). The regions that still cannot be aligned with any short read will be directly discarded, which would compromise the length advantage of long reads. Different from the splitting and trimming strategy, the processing of the HMM-based algorithm is to construct an HMM for an entire long read, update parameters in areas covered by short reads, and maintain parameters previously set based on prior knowledge in areas without short-read coverage. Since HMM cannot capture the long-term dependence between sequences, those uncovered positions are not corrected obviously during the actual error correction process. We hope to be able to capture the correlation between adjacent aligned areas and non-aligned areas and execute reasonable corrections for those uncovered positions. Therefore, RNN, which is good at capturing the correlation between sequences,

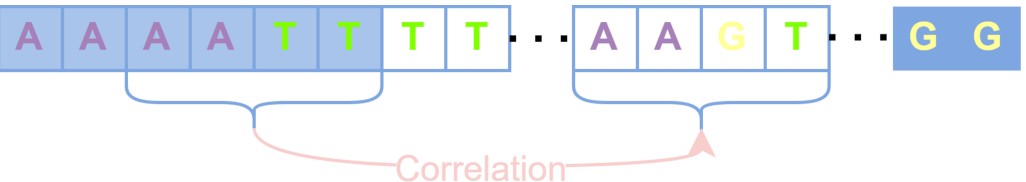

**Figure 3** **The uncovered areas are corrected by long-term dependencies.** Blue squares represent bases covered by short reads, and transparent squares represent bases that are not covered. The correlation among bases "AATT" in the first aligned regions are captured to correct the unaligned bases "AAGT".

is used for error correction, converting the error correction problem into a classification problem in deep learning. How to use the captured long-term dependencies to correct the uncovered long-read regions is shown in Fig. 3. The primary task of classification is the generation of feature vectors and labels.

The generation of feature vectors is according to the principle of majority vote, the more times the current long-read position is covered by a certain type of short-read bases, the greater the probability of that base appearing at that position. Hence, the number of short-read bases aligned to each position is counted based on the generated absolute position, that is, how many times the current long-read position has been aligned by each kind of short-read bases, and then the statistical information is encoded into the feature vector with 11 fields. In the feature vector, fields 1–8 are set as the number of times that the long-read base at the current position is matched with the eight types of bases $"A, T, G, C, a, t, g, c"$ in aligned short reads. If deletions occur at the current position, then at least one short read has no base matched with the base in the current position, and the 9th field marked as "#" isset to 1 to represent deletions occurred at that position; otherwise, this field is set to 0. If the numbers of ambiguous alignment at the current position is greater than the total numbers of short-read alignment, then the 10th field marked by "*" is set to 1 to represent that the position is an ambiguous alignment. The same case can be found at position 21 in Fig. 4. As for the uncovered positions in the long read, there is no auxiliary information for error correction, so the 11th field marked as "SR" represents whether the current position has short-read coverage to assist classification. In this way, the alignment characteristics of each position are encoded into a feature vector composed of 11 fields, and the alignment information is fully utilized. According to the short-read fragments and their alignment relationship with the long-read fragment in Fig. 1, the generation process of several typical pairs of feature vectors and labels is described in Fig. 4.

As for the generation of the label, the base in the short reads that is aligned with a specific position of the long read most frequently is chosen as the ground truth to make the label for that position. That means if a position is covered by short reads, the label is set as the base in the short reads that is the most frequently occurring base at that position. If a long-read position is not covered by any short read the label of the position can only be set to the long-read base itself, and the correction of the uncovered position can only be achieved by the neural network capturing the dependencies between the bases before and after it. Thus, the label character representing the base is mapped to a numerical value according

| Absolute position | Original base | Inserted string*times | Aligned base*times | Feature vetcor [ A, T, G, C, a, t, g, c, #, *, SR] | Label [ A, T, G, C, * ] |
|---|---|---|---|---|---|
| 1 | C | | C*5 | [ 0, 0, 0, 5, 0, 0, 0, 0, 0, 0,1 ] | [ - , - , - , 1 , - ] |
| 2 | A | | A*1,G*1,g*3 | [ 1, 0, 1, 0, 0, 0, 0, 3, 0, 0, 1 ] | [ - , - , 1 , - , - ] |
| 3 | C | | C*5 | [ 0, 0, 0, 5, 0, 0, 0, 0, 0, 0, 1 ] | [ - , - , - , 1 , - ] |
| 4 | N | T*5 | T*5 | [ 0, 5, 0, 0, 0, 0, 0, 0, 0, 0, 1 ] | [ - , 1 , - , - , - ] |
| 5 | A | | G*2,C*2,g*1 | [ 0, 0, 2, 2, 0, 0, 1, 0, 0, 0, 1 ] | [ - , - , 1 , - , - ] |
| ⋮ | | | | | |
| 11 | N | GC*5 | G*5 | [ 0, 0, 5, 0, 0, 0, 0, 0, 0, 0, 1 ] | [ - , - , 1 , - , - ] |
| 12 | N | | C*5 | [ 0, 0, 0, 5, 0, 0, 0, 0, 0, 0, 1 ] | [ - , - , - , 1 , - ] |
| ⋮ | | | | | |
| 19 | N | GC*5, GCC*2 | G*5 | [ 0, 0, 5, 0, 0, 0, 0, 0, 0, 0, 1 ] | [ - , - , 1 , - , - ] |
| 20 | N | | C*5 | [ 0, 0, 0, 5, 0, 0, 0, 0, 0, 0, 1 ] | [ - , - , - , 1 , - ] |
| 21 | N | | C*2,N*3 | [ 0, 0, 0, 2, 0, 0, 0, 0, 0, 1, 1 ] | [ - , - , - , - , 1 ] |
| ⋮ | | | | | |
| 28 | A | | N*5 | [ 0, 0, 0, 0, 0, 0, 0, 0, 1, 0, 1 ] | [ - , - , - , - , 1 ] |
| 29 | A | | N*5 | [ 0, 0, 0, 0, 0, 0, 0, 0, 1, 0, 1 ] | [ - , - , - , - , 1 ] |
| ⋮ | | | | | |
| 43 | A | | Unaligned | [ 1, 0, 0, 0, 0, 0, 0, 0, 0, 1, 0 ] | [ 1 , - , - , - , - ] |

**Figure 4** **The various correspondences between feature vectors and labels.** Several typical examples of the generation of feature vectors and their corresponding labels. At position 1, the long-read base "C" is aligned with short-read base "C" five times, which means the position is covered by short reads (SR field should be set to 1), then the feature vector is encoded as [0, 0, 0, 5, 0, 0, 0, 0, 0, 0,1], and the label is set as "C". The base "T" is inserted five times after position 3, the feature vector for position 4 is encoded as [0, 5, 0, 0, 0, 0, 0, 0, 0, 0, 1], and the label is "T". The bases "GCC" are inserted twice and bases "GC" are inserted three times at position 19, which means base "G" is aligned at position 19 five times, base "C" is aligned at position 20 five times, base "C" is aligned at position 21 twice, base "N" is aligned at position 21 three times. Thus, the feature vector at position 20 is set to [0, 0, 0, 5, 0, 0, 0, 0, 0, 0, 1], and the label is C. Next, the feature vector at position 21 is set to [0, 0, 0, 2, 0, 0, 0, 0, 0, 1, 1], and the 10th field is set to 1, indicating that the position is ambiguous alignment, thus the label is set to *. It should be noted, bases "AA" are deleted at positions 28 and 29 five times, thus, the 9th field "#" of the feature vectors is set to 1. There is no short read aligned with position 43, thus, the 11th field "SR" is set to 0, the 10th field is set to 1 to indicate an ambiguous alignment occurred, and the field "A" should be set to 1 to indicate the original base is "A".

to the rules {"*A*":0,"*T*":1,"*G*":2,"*C*":3,"∗":4}, where "∗" represents an unrecognizable base. The feature vector of each position is a one-dimensional vector of 11 fields, and its corresponding label is a number in {0, 1, 2, 3, 4}. The various correspondences between feature vectors and labels are shown in Fig. 4.

## Model training and base prediction

The error correction task is essentially to modify insertions, deletions, and mismatches in long reads according to the information for the alignment of long reads with high-precision short reads. It can be modeled as an RNN-based multi-class task, and the connection between the feature vector and the target value at each position is obtained by training the neural network to achieve predictive classification. In addition, a long read can be regarded as a time series in which adjacent bases are correlated. Bidirectional RNN (Bi-RNN) is

good at processing sequences with this kind of correlation because of its strong ability to memorize and exploit these pieces of timing information in the sequences and is chosen to perform the correction task in our work. However, the classical Bi-RNN model often suffers from gradient explosion and gradient vanishing in long-range learning. Therefore, as an efficient variant of RNN, bidirectional GRU (Bi-GRU) is selected as the network's input layer to avoid the above problems, and the fully connected layer with a *softmax* activation function is selected as the output layer. The input of the model is the feature vector and its corresponding label for each position. The purpose of model training is to memorize the correspondence between feature vectors and labels and capture the correlation among adjacent bases in the sequences. Error correction is the process of classifying the features of each position through the captured relationships. Bi-GRU cannot only process data from two directions, but each unit in the GRU layer has an update gate and a reset gate, which will capture the dependencies among input vectors in different ranges. Finally, the dependencies and the features are sent to the next *softmax* layer together to calculate for the classification. Our model cannot only capture the correlations among the bases over a long range but also make use of the captured relationships between the feature vector and label pairs to perform classification. This is important for numerous bases with no alignment information available.

In the compilation of the model, the cross-entropy function in multi-class problems is chosen as the loss function, and the back-propagation algorithm is used to update the model parameters. Adaptive moment estimation (Adam) (*Attrapadung et al., 2021*) with an initial learning rate of 0.001 is chosen as the optimizer because it can automatically obtain the step size that needs to be updated by calculating the first- and the second-moment estimates without tuning parameters. During the training of the model, the datasets of feature vectors and labels are split into 80% training data, 10% test data, and 10% validation data. In the prediction of the target values with the model, the feature vectors obtained by encoding the alignment information of the short reads to the long reads to be corrected are sent into the well-trained model for prediction, and the predicted target value representing the base at each position is output. The base sequences are restored according to the mapping between the bases and the target values, and the post corrected reads are restored by concatenating the base sequences according to the corresponding long-read sequence numbers which are saved in the data preprocessing step.

### Network parameter setting

The structure of the Bi-RNN is shown in Fig. 5, $X_1 X_2 \ldots X_{T-1} X_T$ are the feature vectors that are sequentially input to the network, and $Y_1 Y_2 \ldots Y_{T-1} Y_T$ are the corresponding predicted target values. $H_1 H_2 \ldots H_{T-1} H_T$ and $H_1' H_2' \ldots H_{T-1}' H_T'$ are the hidden nodes for forward and reverse connections respectively. If the hidden nodes in the Bi-RNN are replaced by GRU units, a Bi-GRU network can be constructed.

In the internal calculation process of each GRU, it is assumed that the input is $X_t \in \mathbb{R}^{n \times d}$ the hidden state of previous time step is $H_{t-1} \in \mathbb{R}^{n \times h}$, where $n$ is the number of input samples, and $d$ is the number of input features. Then the update gate $Z_t \in \mathbb{R}^{n \times h}$ andreset

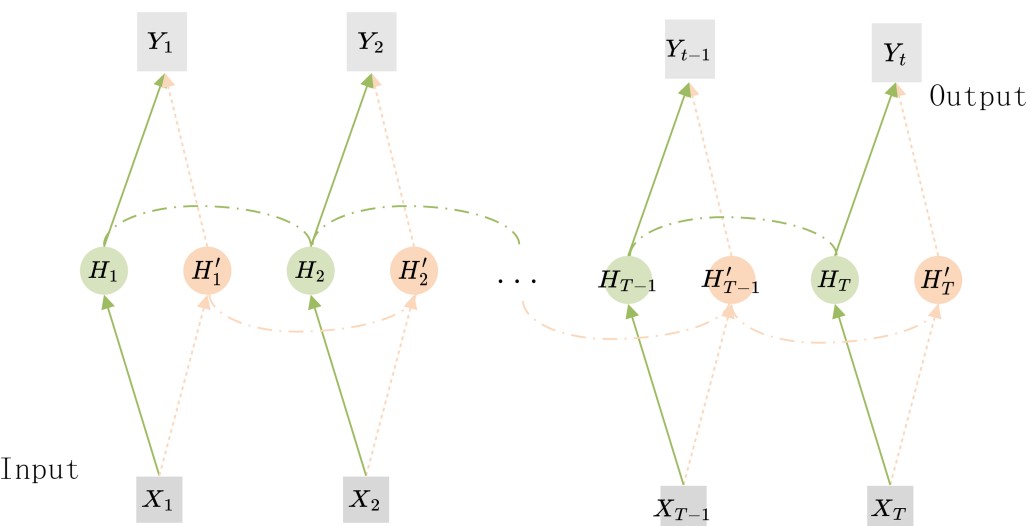

**Figure 5  The structure of the Bi-GRU.** The hidden state information and the output of the current time step depends on the hidden state of its previous and subsequent time steps.

gate $R_t \in \mathbb{R}^{n \times h}$ is calculated with Eqs. (1) and (2):

$$Z_t = \sigma(X_t W_{xz} + H_{t-1} W_{hz} + b_z) \tag{1}$$

$$R_t = \sigma(X_t W_{xr} + H_{t-1} W_{hr} + b_r) \tag{2}$$

where, $W_{xz}, W_{xr} \in \mathbb{R}^{d \times h}$ and $W_{hz}, W_{hr} \in \mathbb{R}^{d \times h}$ are weighting parameters to be updated, $b_z$ and $b_r \in \mathbb{R}^{1 \times h}$ are bias parameter vectors, $h$ is the number of hidden units. $\sigma$ is the activation function *sigmoid*, which is used to map the input value into interval (0,1). Then, Eq. (3) is used to calculate the candidate hidden state $\widetilde{H}_t$ in time step $t$

$$\widetilde{H}_t = \tanh(X_t W_{xh} + (R_t \odot H_{t-1}) W_{hh} + b_h) \tag{3}$$

where, $W_{xh} \in \mathbb{R}^{d \times h}$ and $W_{hh} \in \mathbb{R}^{h \times h}$ are weighting parameters to be updated, $b_h \in \mathbb{R}^{1 \times h}$ is a bias parameter vector. Here, the non-linear activation function tanh isused to ensure that the values of the candidate hidden states are in interval $(-1, 1)$. Then, the obtained candidate state $\widetilde{H}_t$ andupdate gate $Z_t$ are combined to calculate the hidden state of the current time step $H_t$ with Eq. (4)

$$H_t = Z_t \odot H_{t-1} + (1 - Z_t) \odot \widetilde{H}_t \tag{4}$$

.

Finally, the predicted value $\widehat{Y}_t$ iscalculated with Eq. (5).

$$\widehat{Y}_t = \text{softmax}(W_{yh}[H_t, \widetilde{H}_t] + b_y) \tag{5}$$

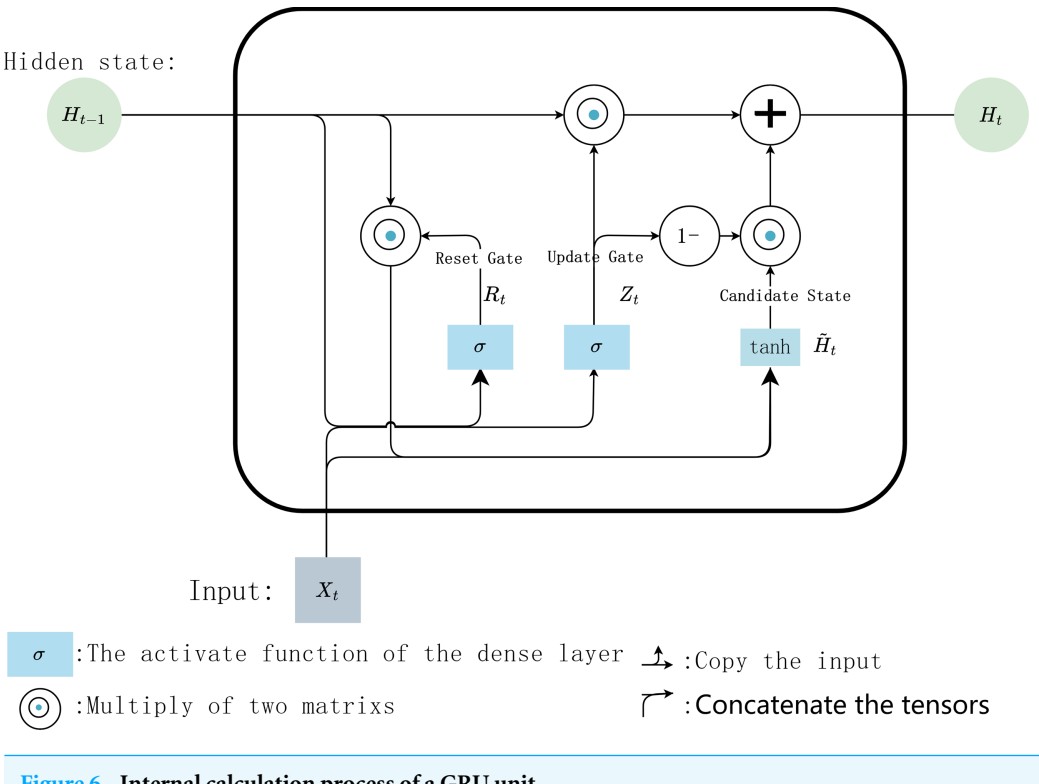

**Figure 6** Internal calculation process of a GRU unit.

where $W_{vh} \in \mathbb{R}^{h \times v}$ are weighting parameters, $b_v \in \mathbb{R}^{h \times 1}$ isa bias parameter vector, $v$ is the number of types of output data. The internal calculation process of each GRU unit is shown in Fig. 6.

Meanwhile, the number of hidden units in each layer is a key parameter. Too many hidden units would lead to overfitting during training, while too few hidden units would miss important features. Based on this consideration, the number of hidden units in the Bi-GRU layer is set to 128*2, and the output of these hidden units will be sent to the following *softmax* layer for classification. There are 5 types of output values at each absolute position. Thus, the number of hidden units in the *softmax* layer is set to 5. The size of the input and output data for each layer of the model is shown in Fig. 7.

## Dataset segmentation

Compared with complex multidimensional data such as image and video data, fewer kinds of features can be extracted for each base position in the long-read sequence, and most of the information that can be extracted is located in the adjacent bases. Therefore, the correlation between adjacent bases in the sequence needs to be carefully preserved for the network when segmenting the training, validation, and test sets. In general, the segmentation of datasets is to randomly split the dataset into disjoint parts according to a certain proportion. However, in our experiments, there are strong correlations among the adjacent bases in each long-read sequence, and the closer the distance, the stronger the correlation. If the training set is randomly segmented as usual, the model will be unable to

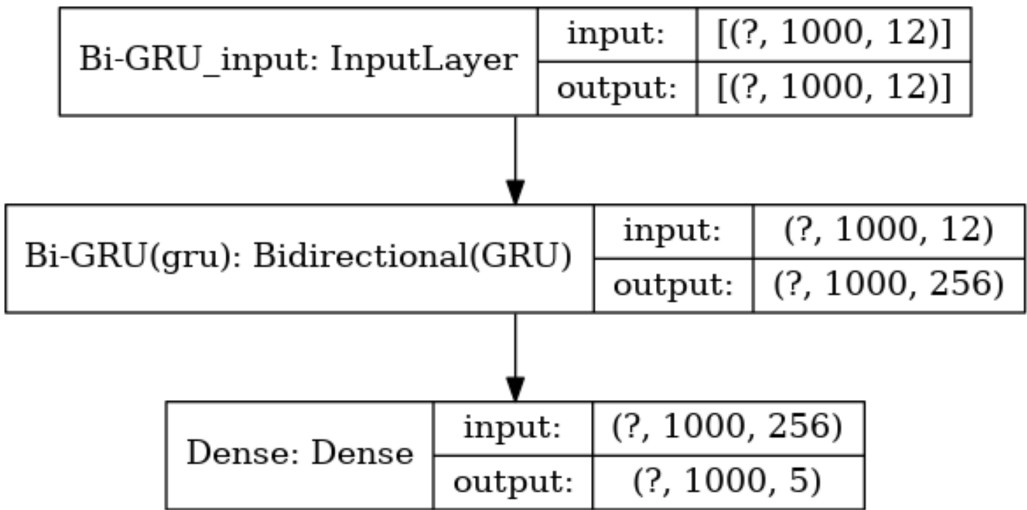

**Figure 7   The size of the input data in each layer.** The size of the input and output data between different layers in the model, where the output of a layer is the input of the next layer. The dimension of the output data of the input layer is represented with (?, 1,000, 12), where "?" is a hyper-parameter referred to as batch-size, which represents the number of samples selected for one training batch. 1,000 represents the length of the current input sequence segment, and 12 is the dimension of each feature vector. The input data dimension for the Bi-GRU layer is (?, 1,000, 256) because the unidirectional GRU has 128 hidden units, and the Bi-GRU layer has a total of 256 hidden units. The output data dimension for the fully connected layer is (?, 1,000, 5), which means that the feature vectors from the input will be classified into 5 categories.

capture the correlation information of distant positions to predict the target value for the current position, which makes the model difficult to fit. The segmentation of training and test datasets should be done on a smaller scale so that the local correlation can benefit the local target prediction. In fact, to make the most use of the correlations among adjacent bases, the specific segmentation consists of two steps: first, the datasets are split into many subsets with a length of 1,000, then the first 80% tensors of each subset are used as the training data, the next 10% are used as the validation data, and the last 10% are used as the test data. The training process of the model with the segmented datasets is shown in Fig. 8.

In the first segmentation step, it should be noted that cutting the sequence data arbitrarily would destroy the continuity of the sequence, resulting in no adjacent relationships available for forward and reverse prediction on the cutting point. Therefore, except for the first subset sequence, the last 200 positions of the previous subset sequence are reserved as the first 200 positions of the current subset sequence. These overlapping regions preserve the continuity of the subset sequences and will be removed according to the same rules during the restoration of the base sequences after the prediction is completed. The training data are used to find the optimal weights of the model, the test data are used to find the best parameters of the optimization algorithm, and the validation data are used to evaluate the trained model. The training process is performed in up to 200 iterations by taking into account the computing power of our computer, and the loss in the back-propagation algorithm, represented as "*Train_loss*", is selected as the indicator to evaluate if the training

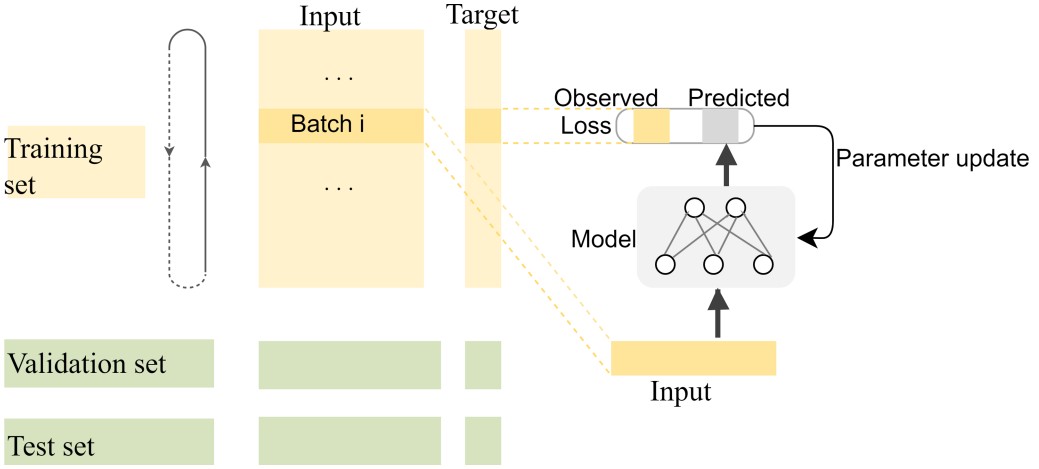

**Figure 8** **The training process of the model with the segmented datasets.** The feature vectors are sent to the neural network in batches for prediction, and the weights of the model are updated by calculating the loss with the predicted values and their corresponding labels. The model is saved for correction until the ideal loss and accuracy of the model are attained.

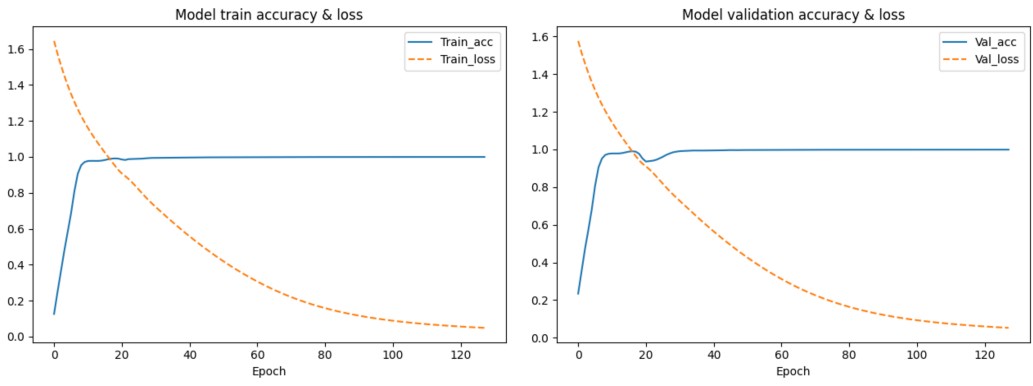

**Figure 9** **The change in the accuracy and loss of the model in training and validation.**

should stop. The validation data is used to test the accuracy and loss of the model after each epoch. If the training accuracy is high but the validation accuracy is low, it means that overfitting has occurred and the hyper parameters should be tuned again. The training will stop if the "*Train_loss*" does not decrease within five epochs. Such an early stopping mechanism can effectively prevent the model from overfitting (*Caruana, Lawrence & Giles Ainips, 2000*). If the training loss does not meet the requirements when the iteration stops, the encoding method of the feature vectors should be modified. When the loss and accuracy no longer change and both reach the ideal value, the iteration will be stopped and the model will be saved for prediction. Finally, the test loss of the model is 0.06026, and the test accuracy is 0.99934. The changes in loss and accuracy of the model along with the increasing of epochs during training and validation are shown in Fig. 9.

**Table 2  Information on the selected datasets.**

| Sequencing specification | Sequencing NCBI accession | Sequencing depth | Number of reads | Reference genome | Genome length(Mbp) | Reference NCBI accession |
|---|---|---|---|---|---|---|
| Illumina Miseq | _ [a] | 373x | 2 × 5,729,470 | *E. coli* K-12 MG1655 | 4.6 | NC_000913.3 |
| PacBio P6C4 | _ [b] | 161x | 87,217 | | | |
| MinION R9 1D | _ [c] | 319x | 164,472 | | | |
| Illumina | ERR1938683 | 81x | 2 × 3,318,467 | *S. cerevisiae* S288c | 12.2 | GCF_000146045.2 |
| PacBio P6C4 | PRJEB7245 | 120x | 239,408 | | | |
| MinION R9 2D | ERP016443 | 59x | 119,955 | | | |
| Illumina | SRX3676782 | 44x | 2 × 20,619,401 | Drosophila melanogaster ISO1 | 143.7 | GCF_000001215.4 |
| Pacbio P5C3 | SRR1204085 | 204x | 6,864,972 | | | |
| MinION R9.5 1D | SRX3676783 | 32x | 663,784 | | | |

**Notes.**
[a] Downloaded from Illumina at Wang, Rongshu (2024). MiSeq_Ecoli_MG1655_110721_PF_R1.fastq. figshare. Dataset. https://doi.org/10.6084/m9.figshare.26021230.v1 and Wang, Rongshu (2024). MiSeq_Ecoli_MG1655_110721_PF_R2.fastq. figshare. Dataset. https://doi.org/10.6084/m9.figshare.26021290.v1.
[b] Downloaded from PacBio at https://github.com/PacificBiosciences/DevNet/wiki/E.-coli-Bacterial-Assembly
[c] Downloaded from Loman Labs at https://s3.climb.ac.uk/nanopore/E_coli_K12_1D_R9.2_SpotON_2.pass.fasta

## Dealing with GPU overflow

In addition to the huge amount of long-read data itself, the datasets of feature vectors and labels generated in preprocessing are also huge. As an example, a 730 MB long-read dataset can generate 76 GB datasets of feature vectors and labels after alignment and data preprocessing. Since the neural network is deployed on the Graphic Processing Unit (GPU) for training and prediction, GPU not only needs to save many model parameters and training data but also needs to perform many nonlinear computations. If all feature vectors and labels are fed into the network, the memory of the GPU would overflow; thus, it is necessary to build a suitable data generator to send the data to the GPU in batches for processing. In this experiment, the Sequential class in the library of Keras is inherited to build our own data generator. When generating data in batches, the initially segmented subset of feature vectors and labels will be further split into training data and test data. Finally, batches of generated training data, test data, and validation data are fed into the network for training and prediction. In this way, memory overflow of the GPU can be effectively avoided.

## RESULTS AND DISCUSSION

### Datasets and experiment setup

The six datasets used in our experiments are sequenced from three species, *Escherichia coli* K-12 MG1655 (*E. coli*), Saccharomyces cerevisiae S288C (yeast), and Drosophila melanogaster ISO1 (fruit fly). For each genome, the long reads to be corrected are from the PacBio and Nanopore platforms, and the high-quality short reads that are used to correct long reads are from the Illumina platform. *E. coli* K-12 MG1655, S. cerevisiae S288c, and Drosophila melanogaster ISO1 are the reference genomes for evaluating the correction quality, and they are sequenced and assembled carefully by Sanger and other institutions. The details of the datasets are listed in Table 2.

DeepCorr and five other typical short reads-involving error correction algorithms are used to correct the above datasets To provide research insights in the absence of short reads, we also included the latest self-correction algorithm VeChat for comparison. All experiments in this work are run on a server (dual Intel Xeon Gold 6240 @ 2.60 GHz) with 72 cores, 256 GB Central Processing Unit (CPU) memory, and 2 GPUs (Quadro RTX 6000, Compute Capability 7.5). The two GPUs are combined to construct one GPU with larger memory. The compared algorithms LoRDEC, Jabba, ColorMap, Proovread, and Hercules are run with the default parameters on the CPU cores of the server using multithread processing. The command line parameters of algorithms involved in this work are based on the manual provided by respective authors, and the details are recorded in supplementary file. For DeepCorr, the process of alignment and the generation of feature vectors and labels are implemented on the CPU, and the training and prediction of the RNN model are implemented on the GPU. All source codes are based on Python 3.6 and TensorFlow-gpu 2.3.

## Accuracy metrics

As described by LRECE (*Zhang, Jain & Aluru, 2020*), although the error correction algorithm lacks the ground truth, differences between the corrected long reads and the reference genome mean uncorrected errors. The reference genome is the homologous high-precision genome sequence obtained by authoritative institutions combining various high-quality sequencing methods at a great cost. In this way, the quality of error correction can be obtained by evaluating the quality of the alignment of the corrected sequence to the reference genome. In practice, *Minimap2* is used to align both the original and the corrected long reads to their reference genome. Finally, various performance indicators of these alignments are calculated to evaluate the error correction performance of the algorithms. The raw experimental results are calculated by LRECE. The original results on six benchmark datasets are shown in Tables S1–S6.

In the experimental results, "Total bases" is the total number of bases of the long read after correction, the difference on this indicator between the corrected long read and the original one should not be too large. "Aligned bases" is the number of corrected bases that can be aligned to the reference genome. "Alignment identity" represents the consistency of the segments in the long reads and the corresponding aligned fragments in the reference genome, which is calculated by dividing "Aligned bases" by the "Total bases". Obviously, increasing total bases will lead to a decrease in alignment identity under the same aligned bases. In terms of DNA data processing, if the read is long enough, there is no need for polymerase chain reaction (PCR) amplification, which can avoid base bias and simplify genome assembly. Thus, we also compared the length of long reads after correction. "Maximum length (bp)" and "Average length (bp)" are the maximum and average lengths of regions where long reads can be aligned to the reference genome respectively. "N50″is a length indicator. All long reads are sorted in the order of their lengths and concatenated into a sequence, and then, the midpoint of this sequence is found. The length of the long read where this midpoint is located is the N50 indicator. "Genome fraction (%)" is the ratio of the number of bases covered by the long-read assembly in the genome divided by

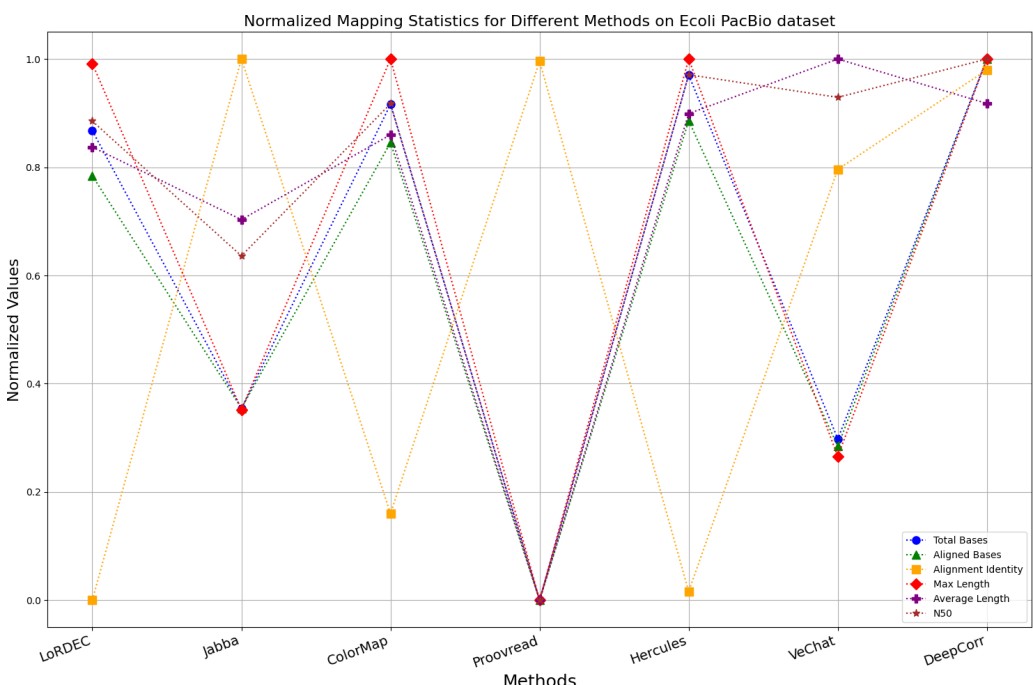

**Figure 10 Normalized metrics on the *E. coli* PacBio dataset.**

the total length of the reference genome. N50 and genome fraction are both metrics that reflect sequence continuity from an assembly perspective. "Memory usage (GiB)" is the peak CPU memory occupied by each algorithm during the correction.

The evaluation metrics for error correction tasks encompass multiple aspects, often requiring considerations of trade-offs and compromises. Therefore, making a direct judgment on the superiority or inferiority of a certain method is often challenging. To visually present the performance of each method on the same chart, we have employed Min-Max normalization to process the metrics except for the time and the memory requirement, aiming to eliminate the scale differences among various indicators. The formula for this normalization process is as follows:

$$X_{normalized} = \frac{X - X_{\mathbf{min}}}{X_{\mathbf{max}} - X_{\mathbf{min}}} \qquad (6)$$

Where, $X_{Normalized}$ is the normalized value, $X$ is the original data value, $X_{\mathbf{min}}$ is the minimum value of the original data, $X_{\mathbf{max}}$ is the maximum value of the original data. According to the normalization principle, algorithms with better performance in various metrics will have markers concentrated near 1, while algorithms with poor performance will have markers clustered near 0.

It is worth noting that if a certain metric of the algorithm deviates from 1, it indicates that the algorithm involves significant trade-offs on this metric. Finally, the data presentation of the raw results based on six benchmark data sets of data is provided in the supplementary file, and the intuitively normalized display results are shown in Figs. 10–15.

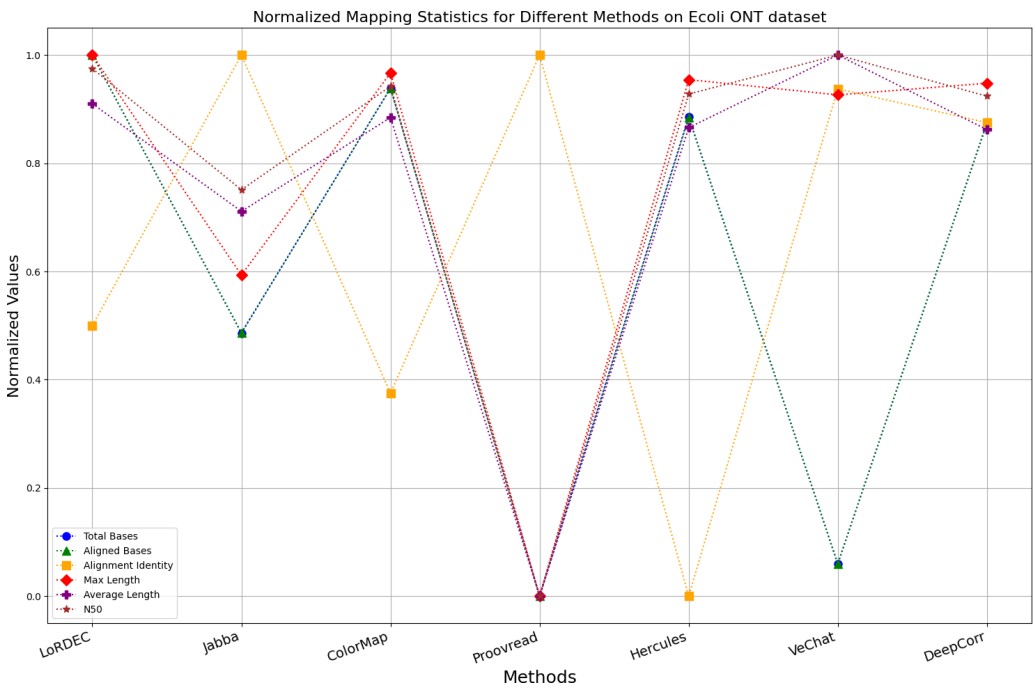

**Figure 11** **Normalized metrics on the *E. coli* ONT dataset.**

## DISCUSSION

### Analysis of the results

Enhancing the alignment consistency between long reads and the reference genome is a fundamental objective in error correction. However, achieving this goal needs to consider a comprehensive array of metrics. Figures 10, 11, 12, 13, 14 and 15 illustrate the normalized metrics across each benchmark dataset. The graphical representation reveals that DeepCorr demonstrates a notably tighter clustering around 1 across five of the six datasets except the fruit fly PacBio dataset. This observation suggests its superior performance with minimal discernible weaknesses. The following discussion delves into the comparative outcomes between DeepCorr and the remaining six methods.

### Trimming consideration

During error correction, there should be no significant loss of bases. Therefore, we first assess the feasibility of algorithms with trimming strategies. Jabba demonstrates a remarkable alignment identity metric ranging from 0.99 to 1.00 in most instances, primarily attributed to its trimming approach. Specifically, Jabba trims the extended extremities of long reads that extend beyond the paths in the constructed De Bruijn graph, thus enhancing alignment identity metrics. However, this trimming strategy leads to substantially smaller values for metrics such as lengths and N50 compared with other algorithms, resulting in the loss of global information and the inherent length advantages of long reads. Additionally, the long-read file corrected by Jabba is reduced to one-third of its original size. Coincidentally, a similar situation also observed with Proovread

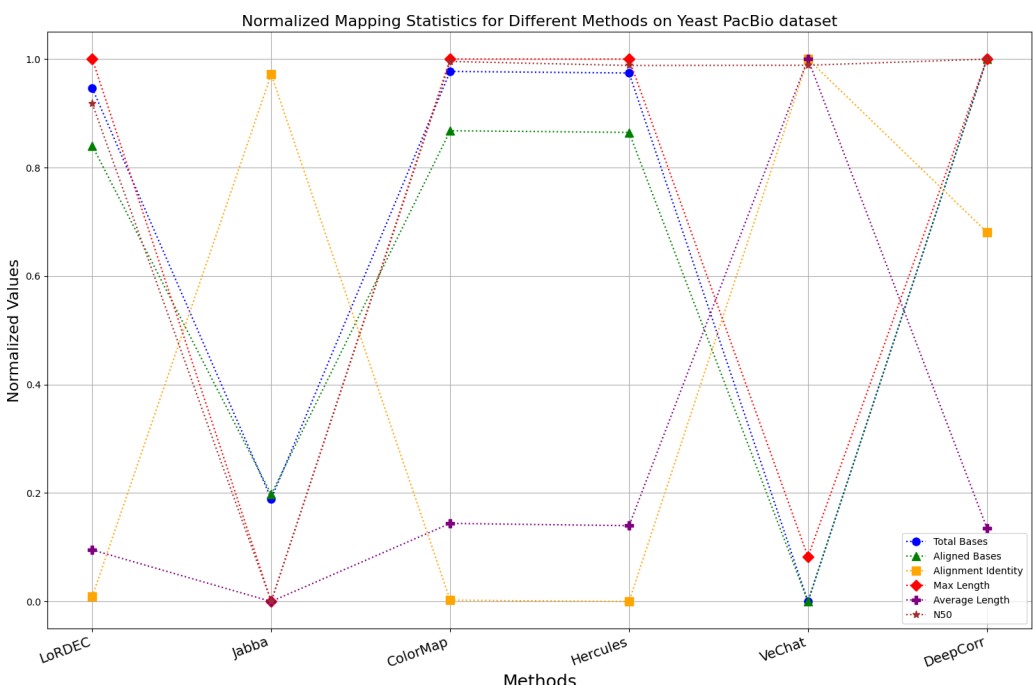

**Figure 12  Normalized metrics on the Yeast PacBio dataset.**

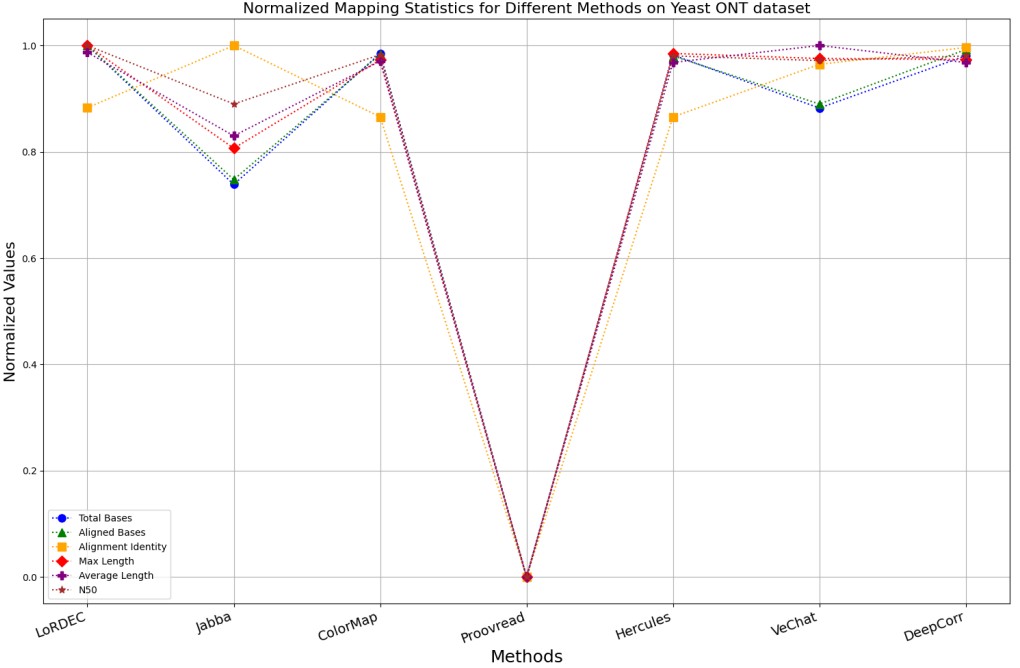

**Figure 13  Normalized metrics on the Yeast ONT dataset.**

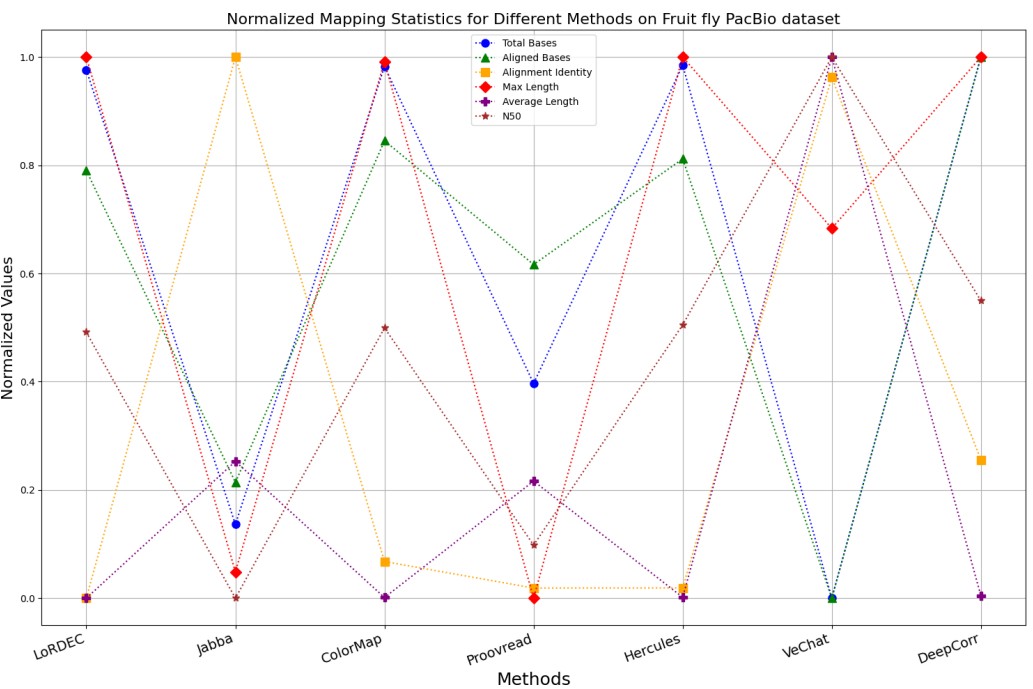

**Figure 14  Normalized metrics on fruit fly PacBio dataset.**

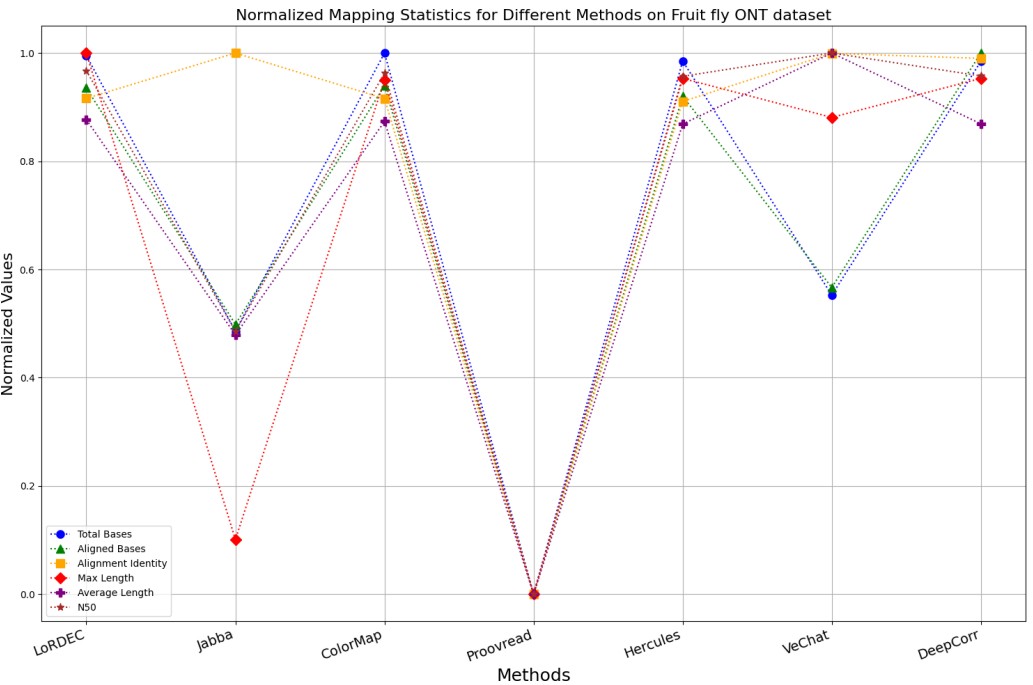

**Figure 15  Normalized metrics on fruit fly ONT dataset.**

and VeChat. Apart from Fig. 13, the number of corrected long-read bases by VeChat significantly decreases, attributed to its selection of the most frequent nodes (*i.e.,* highest accuracy paths) during path planning, thereby sacrificing the longest paths. Long-read fragments and reads that cannot be mapped to the partial order alignment (POA) graph are disregarded. Therefore, while the three algorithms mentioned above demonstrate high accuracy, they sacrifice other metrics such as total base count and the maximum length. Consequently, they exhibit increased alignment identity but noticeable trade-offs in other metrics, shown as more dispersed metrics of total bases and the maximum read length on the normalized figure. Researchers conducting subsequent analyses should exercise caution when employing these methods.

### DBG-based and alignment-based algorithms

Figs. 11 and 13 illustrate LoRDEC's superior performance in metrics such as aligned base count and alignment identity, as well as its advantages in length indicators and N50. However, Figs. 10, 12, 14 and 8 demonstrate DeepCorr's superiority over LoRDEC from more compact metrics' layout. The discrepancy in each metric between these two methods is less than 5%. Thus, it can be inferred that both methods exhibit similar error correction performance. Similar to other algorithms relying on k-mer frequencies, LoRDEC's performance is heavily influenced by parameter configurations. Hence, selecting appropriate parameters and optimal paths based on experience is crucial for data obtained from diverse sequencing platforms and species. When the value of k is too small, it may lead to an excessive number of optimal paths, increasing computational resources. Conversely, when the value of k is too large, it may miss the optimal path. This requires researchers to have sufficient experience of the error characteristics in the sequencing data in order to set the optimal parameters. Figures 14 and 15 showcase DeepCorr's exceptional performance in terms of aligned base count and alignment identity, indicating its ability to handle data with more intricate structures, such as fruit fly data. Alignment information is the sole requirement for generating feature vectors and labels, irrespective of the complexity of the structure. However, DeepCorr consumes less user time compared with LoRDEC.

For the alignment-based method, DeepCorr not only performs better than ColorMap in terms of the total number of bases after correction and alignment identity, but also performs well in length indicators in Figs. 10–14. Looking at the overall situation, DeepCorr outperforms both alignment-based methods in all indicators.

### The only machine learning-based algorithm

Based on observations of corrected long reads from Hercules, it is found that it only corrects regions with short read alignments, while regions without short read alignments receive no change. There are two benefits to this approach: (1) preserving the length advantages and global information of long reads, and (2) automatically adjusting regions covered by short reads appropriately ignoring the different error profiles from the two mainstream sequencing platforms. The only weakness is that Hercules is subjected to the limitation of the order of the HMM, which assumes that the current base is only related to the previous base, which prevents the model from capturing longer-term dependencies between bases

in each long read for error correction of unaligned bases. Therefore, RNNs are utilized as a novel model for time-series data in DeepCorr, which assumes that the current base is correlated with all previously occurred bases, enabling it to capture longer dependencies. Experimental results demonstrate that, in addition to achieving the two breakthroughs accomplished by Hercules, DeepCorr also increases the number of bases aligned to the reference genome after error correction and the alignment identity, indicating that its correction process automatically converges towards the expected direction (the reference genome) without human intervention.

## A set of special data

For the fruit fly dataset from PacBio platform, as shown in Fig. 14, since there is an obvious difference between the original long-read data and the reference genome, the alignment identity is quite low. Although all methods try their best to correct errors, the results are unsatisfactory. In this set of results, Jabba, Proovread and VeChat exhibit a peculiar doubling of average length, despite a significant decreasing in total bases and maximum length. The reason for this anomaly is that many shorter and low-quality long reads that failed to align to high-accuracy short reads or to the De Bruijn graph (DBG) constructed from these short reads have been removed by these two methods. However, other algorithms retain these regions, which result in suboptimal alignment identity. Nonetheless, DeepCorr reports the highest alignment identity among the other methods while preserving the total base count and the maximum length.

## Resource consumption statistics

### Memory usage

For resource consumption statistics, the training and prediction processes of DeepCorr are executed on the GPU, and the processes of alignment and feature generation are run on the CPU. The generated feature and label datasets are efficiently stored in the CPU in the form of HDF5, waiting to be sent into the GPU in batches for training. In terms of memory usage, both GPU memory and main memory are used. Although DeepCorr only consumes 3.4 GiB of main memory, the training and prediction process of the model consumes 23190 MiB of GPU memory. Other compared algorithms only consume main memory, and their memory consumption is shown in the Supplementary files. VeChat consumes the most amount of memory (more than 50GiB in the smallest dataset).

## Time complexity

Since the core layer of the model is a single RNN layer, the time complexity of a single time step in an RNN model is typically denoted as O (N), where N represents the dimensionality of the input data. This complexity arises from the computations performed at each time step, including matrix multiplications, element-wise operations, and activation functions applied to the input and hidden states. The computational complexity for every time step remains constant regardless of the sequence length. However, when considering the entire genome sequence, the time complexity is proportional to the sequence length due to the sequential nature of RNN processing, resulting in a total time complexity of O (T), where T is the length of the sequence.

To evaluate the computational power of the algorithms, The Unix "time" command is used to record the computational power of each method, then the command line will output "real time", "sys time" and "user time", where "user time" is all the time spent in user space during program execution, and it does not change with the number of cores in the CPU. "User time" is chosen as the indicator in the results. In Tables S2–S7 of the supplementary file, Jabba is the fastest, but this is obtained by sacrificing the length of long reads seriously. Moreover, DeepCorr still consumes less user time even if dealing with fruit fly data with more complex structures, which can be seen in Tables S6–S7 of supplementary file. It is worth noting that before the model starts training and prediction, those long reads uncovered by any short read will not be processed further, and only the long reads covered by at least one short read will be encoded and input for training and prediction. In fact, it takes 124 min of user time to align 2.1 GiB short reads to 0.7 GiB *E. coli* long reads and 526 min to train and predict the result, and 569,261 k bases involved in training and prediction, thus the training and prediction time used for every 1,000 k bases is 55.44 s. It takes 267 min of user time to align 1 GiB short reads to 5.5 GiB yeast long reads and 1,926 min to train and predict the result, and 2,451,999 k bases involved in training and prediction, so the training and prediction time required for every 1,000 k bases is 47.13 s. It takes 4,760 min user time to align 4 GiB short reads to 4.5 GiB fruit fly long reads and 2,871 min to train and predict, and 3,375,564 k bases involved in training and prediction, and it takes 50.02 s for every 1,000 k bases, which is almost the same as the previous two results. It can be concluded that computational complexity of DeepCorr during the training and prediction is only related to the number of bases involved, which is roughly linearly related to the number of long reads bases, and which is not related to the complexity of the structure of long reads.

## CONCLUSIONS

The length advantage of long read data sequenced by third-generation sequencing techniques is limited by the high error rate. Generally, homologous high-precision short reads are effective for long-read error correction. However, the uncovered long-read regions cannot be corrected easily by the previous methods. In this work, a sequencing technology-independent error correction method is transformed into a multi-class problem in neural network to capture the long-term dependencies among bases, which not only performs accurate error correction on aligned regions, but also takes good care of the rest unaligned regions, which can be inferred from its maximum number of aligned bases and highest alignment identity.

In conclusion, the major advantage of DeepCorr is the highest alignment identity and number of aligned bases, even on the more complex structure datasets such as fruit fly datasets, so that it can produce more accurate and longer reads. The best part is that DeepCorr can correct bases that do not have any coverage by the captured long-term dependencies. Moreover, none of the bases is trimmed off from the long reads to maintain the read length advantage. In terms of computing resource consumption, DeepCorr consumes significantly less user time and is suitable for long reads from both PacBio and

ONT platforms. This is the only GPU-based algorithm. With the rapid development of GPU computing power, the speed of this algorithm will also increase, which can provide a new perspective for solving long-read error correction with deep learning. In the future, how to build a generative pretrained model to automatically extract features from alignment information for producing accurate long reads will be studied.

### Funding

This work was funded by the National Natural Science Foundation of China under Grant (61861045). The funders had no role in study design, data collection and analysis, decision to publish, or preparation of the manuscript.

### Grant Disclosures

The following grant information was disclosed by the authors:
National Natural Science Foundation of China: 61861045.

### Competing Interests

The authors declare there are no competing interests.

### Author Contributions

- Rongshu Wang conceived and designed the experiments, performed the experiments, analyzed the data, performed the computation work, prepared figures and/or tables, authored or reviewed drafts of the article, and approved the final draft.
- Jianhua Chen conceived and designed the experiments, analyzed the data, authored or reviewed drafts of the article, and approved the final draft.

### Data Availability

The sequences are available at NCBI: ERR1938683, PRJEB7245, ERP016443, SRX3676782, SRR1204085, SRX3676783.
The data is available in the Supplemental File.

### Supplemental Information

Supplemental information for this article can be found online at http://dx.doi.org/10.7717/peerj-cs.2160#supplemental-information.

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
