# Peer review of "DeepCorr: a novel error correction method for 3GS long reads based on deep learning"

_PeerJ Computer Science, doi:10.7717/peerj-cs.2160_

## Round 0.1 · original submission · Major Revisions

The reviewers have substantial concerns about this manuscript. The authors should provide point-to-point responses to address all the concerns and provide a revised manuscript with the revised parts being marked in different color.

Reviewer 1 ·

Basic reporting

no comment

Experimental design

no comment

Validity of the findings

no comment

Additional comments

no comment

Annotated reviews are not available for download in order to protect the identity of reviewers who chose to remain anonymous.

Reviewer 2 ·

Basic reporting

This study proposed an interesting application of neural network models on long read sequencing technology, with special focus on improving the error correction using high-precision short reads. The authors benchmarked their method against previously published methods including short-read-DBG based methods and short-read-alignment-based methods, also HMM-based methods. However, due to the writing, the entire manuscript was extremely challenging to read through and the methodology was hard to follow given too much unnecessary technical details and messy logic flow, I found it has a long way to go before becoming publishable. Therefore, given the limit understanding got from this manuscript, I could only provide a few high-level comments as follows:
1. The training datasets seemed to be limited to three small genomes of bacteria, yeast and drosophila, and the short reads seemed to have poor coverage when mapped to long reads (range from 0.02 – 0.36), which made the quality of training data (and testing data) seemed less convincing. Meanwhile, the uncovered bases could be as high as 98% -- based on my personal experience working with both long reads and short reads data, this shouldn’t happen if the sequencing depth is decent (a missing information from the manuscript).
2. Given the high computational cost and the fact that one would have to re-sequence the same samples using deep short reads sequencing, I don’t think this method is cost-effective in general genetic research, except very limited application of small genome with highly repetitive structure that required long-reads technology, and for some reason the precision of the sequencing is highly required. For large genome like human samples, I’m not sure how useful this method could be.
3. The performance of this method also varied a lot (table 3-8), and I’m wondering if the authors had considered some systematic evaluation of the factors that affect the performance (e.g. time complexity, accuracy, pattern of misclassified regions), such as the length of genome, the complexity of the sequences, short-reads coverage etc.
4. There are way too many tables and figures and lots of unnecessary details in the main text. I would recommend combining + reducing the main figures/tables to maximum of four, highlighting what were the most important messages and put the rest of the information into supplementary.

Experimental design

Given the high computational cost and the fact that one would have to re-sequence the same samples using deep short reads sequencing, I don’t think this method is cost-effective in general genetic research, except very limited application of small genome with highly repetitive structure that required long-reads technology, and for some reason the precision of the sequencing is highly required. For large genome like human samples, I’m not sure how useful this method could be.

Validity of the findings

The training datasets seemed to be limited to three small genomes of bacteria, yeast and drosophila, and the short reads seemed to have poor coverage when mapped to long reads (range from 0.02 – 0.36), which made the quality of training data (and testing data) seemed less convincing. Meanwhile, the uncovered bases could be as high as 98% -- based on my personal experience working with both long reads and short reads data, this shouldn’t happen if the sequencing depth is decent (a missing information from the manuscript).

Reviewer 3 ·

Basic reporting

1. Although the correction of long-read sequencing data remains an active area of research, the references in this article to the concerned long-read sequencing technology seem incomplete. For instance, the manuscript states that HiFi sequencing reads are typically 1-2 kb (lines 10-12). However, this does not align with the more recent specifications (~15 kb) for HiFi sequencing technology, as reported by the company and other researchers. To maintain accuracy and relevance, please update the information to reflect the latest technology and provide a more comprehensive review of current long-read sequencing capabilities.
2. The legend for Figure 4 appears to contain some confusion, as the positions indexed as 19/20/21 in the figure seem to disagree with the text description. Please review and correct any inconsistencies between the figure and the corresponding text.

Experimental design

1. The base correction software reviewed and referenced in the manuscript were all published before 2020. However, several newer tools have been released in recent years, such as NextDenovo, CONSENT, and VeChat. The absence of these more recent references raises questions about the claimed knowledge gap and whether the current study effectively addresses the latest developments in this field. It would be helpful for the authors to include a more comprehensive review of recent base correction software to accurately reflect the state-of-the-art and better justify the significance of the current study.

Validity of the findings

1. Because no similar tools published after 2020 were compared, the manuscript's claim of improved performance of the newly developed tool against older competing tools may lack sufficient evidence. Although the results suggest a clear advantage over earlier base correction tools, the absence of comparisons with more recent software leaves the claimed improvement relative to the current state-of-the-art in question. To strengthen the manuscript and justify the contribution to the field, the authors should consider including comparisons with tools published after 2020 or discussing why these comparisons were not feasible.

Additional comments

1. The method presented in the manuscript relies on high-quality short-read data for correcting long-read sequences. This requirement could be considered a limitation, as it may restrict the use of this approach in contexts where high-quality short-read data is unavailable or difficult to obtain. It would be beneficial for the authors to discuss this limitation in the manuscript, including the implications for practical application and possible workarounds or alternative methods for scenarios where short-read data is not readily accessible.
2. It's well acknowledged that long-read sequencing technologies are more prone to errors when sequencing repetitive regions, which should agree largely with the regions where short reads are poor aligned due to limited read length. Since the new approach is shown to have superior performance in correcting bases not aligned by high-quality short reads, it would be helpful and important for the authors to delve deeper into these improved regions. By examining these regions, the authors could provide a better explanation of how the new approach achieves its superior performance. This deeper analysis could offer insights into the model's effectiveness and help validate the method's robustness, especially in challenging sequencing regions.

---

## Round 0.2 · accepted · Accept

The reviewers are satisfied with the revisions, and I concur to recommend accepting this manuscript.

Reviewer 1 ·

Basic reporting

I apologize for submitting an incorrect version of my comments; however, I have reviewed the response to another reviewer, which addresses my major concerns as well. The author has already addressed all of my concerns, and I believe the manuscript can be published with minor grammatical and formatting modifications

Experimental design

no comment

Validity of the findings

no comment

Additional comments

no comment

Reviewer 3 ·

Basic reporting

All of my comments have been addressed by the authors.

Experimental design

All of my comments have been addressed by the authors.

Validity of the findings

All of my comments have been addressed by the authors.

Additional comments

All of my comments have been addressed by the authors.